# Music to prevent deliriUm during neuroSurgerY (MUSYC): a single-centre, prospective randomised controlled trial

Pablo R Kappen ,[1] M I Mos,[1] Johannes Jeekel,[1] Clemens M F Dirven,[1] Steven A Kushner,[2] Robert-Jan Osse,[2] Michiel Coesmans,[2] Marten J Poley,[3,4] Mathijs S van Schie ,[5] Bronno van der Holt,[6] M Klimek ,[7] Arnaud J P E Vincent[1]

For numbered affiliations see end of article.

**Correspondence to**
Dr Pablo R Kappen;
p.r.kappen-2@umcutrecht.nl

## ABSTRACT

**Objectives** Delirium is a serious complication following neurosurgical procedures. We hypothesise that the beneficial effect of music on a combination of delirium-eliciting factors might reduce delirium incidence following neurosurgery and subsequently improve clinical outcomes.

**Design** Prospective randomised controlled trial.

**Setting** Single centre, conducted at the neurosurgical department of the Erasmus Medical Center, Rotterdam, the Netherlands.

**Participants** Adult patients undergoing craniotomy were eligible.

**Interventions** Patients in the intervention group received preferred recorded music before, during and after the operation until day 3 after surgery. Patients in the control group were treated according to standard of clinical care.

**Primary and secondary outcome measures** Primary outcome was presence or absence of postoperative delirium within the first 5 postoperative days measured with the Delirium Observation Screening Scale (DOSS) and, in case of a daily mean score of 3 or higher, a psychiatric evaluation with the latest Diagnostic and Statistical Manual of Mental Disorders (DSM-5) criteria. Secondary outcomes included anxiety, heart rate variability (HRV), depth of anaesthesia, delirium severity and duration, postoperative complications, length of stay and location of discharge.

**Results** We enrolled 189 patients (music=95, control=94) from July 2020 through September 2021. Delirium, as assessed by the DOSS, was less common in the music (n=11, 11.6%) than in the control group (n=21, 22.3%, OR:0.49, p=0.048). However, after DSM-5 confirmation, differences in delirium were not significant (4.2% vs 7.4%, OR:0.47, p=0.342). Moreover, music increased the HRV (root mean square of successive differences between normal heartbeats, p=0.012). All other secondary outcomes were not different between groups.

**Conclusion** Our results support the efficacy of music in reducing the incidence of delirium after craniotomy, as found with DOSS but not after DSM-5 confirmation, substantiated by the effect of music on preoperative autonomic tone. Delirium screening tools should be validated and the long-term implications should be evaluated after craniotomy.

**Trial registration number** Trialregister.nl: NL8503 and ClinicalTrials.gov: NCT04649450.

## STRENGTHS AND LIMITATIONS OF THIS STUDY

⇒ This is the first randomised controlled trial assessing the effect of music on delirium after craniotomy.
⇒ A variety of secondary outcomes, substantiating the onset of delirium and its clinical implications, were collected.
⇒ Delirium was defined with the Delirium Observation Screening Scale and the Diagnostic and Statistical Manual of Mental Disorders criteria.
⇒ Due to the nature of the intervention, we did not blind the study, which could have influenced the outcome assessors.
⇒ The generalisability of the results may be affected by the single-centre design of the study.

## INTRODUCTION

Delirium is defined in the latest Diagnostic and Statistical Manual of Mental Disorders (DSM-5) as 'an acute disturbance in attention and cognition which is not better explained by another neurocognitive disorder such as for example dementia'. To increase the recognition of delirium during hospital stay, a variety of delirium diagnostic screening tools have been developed, which can also be assessed by other healthcare workers. Delirium in neurosurgical patients has been reported in 4%–44% of cases, with a large variation in definition and assessment methods.[1] The high incidence in this population is probably caused by the underlying massive neuroinflammation which is usually induced during intracranial procedures.[2] Delirium, also in neurosurgical literature, is often multifactorial in aetiology and can be influenced by a number of predisposing (eg, older age, cognitive impairment, multiple comorbidities) and precipitating factors (eg, infections, operations, drugs).[3–8] The clinical relevance of delirium in neurosurgery remains difficult to assess objectively, as criteria for delirium overlap with symptoms from the primary

neurological injury. However, delirium independently predicted clinical outcomes in neurosurgical and neurocritically ill patients such as worse functional outcome,[9] length of stay, costs and death.[10] These complications justify the search for preventive therapies for postoperative delirium in neurosurgical patients.

Although promising preventive approaches are emerging, pharmacological treatments have inconsistent results and are accompanied with side effects.[11 12] Non-pharmacological multicomponent approaches for primary prevention, such as reorientation, early mobilisation, therapeutic activities, hydration, nutrition and sleep strategies, have been shown to be effective and cost-reducing in other patient groups. However, these approaches can be labour intensive, and include the use of volunteers or non-licensed professionals to enhance feasibility.[13]

Recorded music is an easy applicable intervention which neatly fits throughout the entire perioperative process and has been shown to be effective in the surgical population in reducing a combination of delirium-eliciting factors such as preoperative anxiety, postoperative pain, stress response and opioid/sedation requirement.[14–21] A recent meta-analysis, evaluating six randomised pilot studies, found music potentially being effective in preventing postoperative delirium in postsurgical patients. However, these studies did not include neurosurgical patients.[22]

We therefore designed a randomised controlled trial to assess the effect of music on the prevention of postoperative delirium in neurosurgical patients.

## METHODS
### Patient and public involvement
Patients were involved in the composition of the music playlists, as these were based on their music preference, the role music plays in their life (ie, whether they are musician/just listen to music) and the importance of music. The results of our trial were disseminated to the participating patients through a letter after publication.

### Study design
The Music to prevent deliriUm during neuroSurgerY Clinical (MUSYC) trial was a single-centre, prospective randomised controlled trial conducted at the Erasmus Medical Center (MC), Rotterdam, the Netherlands. The trial compared effects of music administered before, during and after craniotomy with standard of clinical care.

The trial protocol was designed by neurosurgeons, psychiatrists, anaesthesiologists and neuroscientists and followed the Standard Protocol Items: Recommendations for Interventional Trials guidelines and the Consolidated Standards of Reporting Trials guidelines for non-pharmacological treatments (see checklist in online supplemental material). The trial was registered (trialregister.nl: NL8503 and ClinicalTrials.gov: NCT04649450)

and details of the protocol have been published previously.[23 24]

We expected an incidence of delirium in our control group of 30%, which was based on the incidence of 24.2%–32.4% documented in neurosurgical literature using the same screening tool (ie, Delirium Observation Screening Scale (DOSS)).[5 7] When designing the trial, the expected effect could not be based on previous literature as no pooled effect of music on delirium was reported. Other non-pharmacological interventions mentioned a relative reduction of 36%–77% and we therefore considered the intervention clinically relevant if a relative reduction of 60%, corresponding to an absolute reduction of 18%, was achieved.[25 26] Assuming a loss to follow-up of 5%, we estimated that a target sample size of 189 patients would provide the trial with a power of 80%.

From July 2020 through September 2021, 189 patients were registered and randomly assigned to a trial group: 95 in the music group and 94 in the standard care group (figure 1). Randomisation was done in a 1:1 ratio, by a secured online software program (ALEA; FormsVision, Abcoude, the Netherlands) and stratified per type of disease characteristic (ie, 'neuro-oncology', 'neurovascular', 'traumatic brain injury', 'infectious') and age (ie, 'younger than 60 years', '60 years or older'). Variable block sizes were used in which in each block, both groups were represented equally.

### Patients
Adult patients (ie, age 18 years or more) undergoing craniotomy (ie, opening the dura requiring bone flap removal) at the Erasmus MC with sufficient knowledge of the Dutch language were eligible for study participation. Exclusion criteria were: impaired awareness before surgery (ie, motoric less than 6 in the Glasgow Coma Scale), planned postoperative intensive care unit (ICU) admission, suspected delirium (defined as fluctuating awareness) at baseline, antipsychotic treatment, undergoing surgery impeding supply of music (ie, awake craniotomy or vestibular schwannoma surgery), bilateral hearing impairment and participation in other clinical trials interfering with results. During inclusion, one participant reported that music induced epileptic seizures (known as musicogenic epilepsy): this patient was excluded (and the exclusion criteria were adopted accordingly), as it was considered unethical to expose such a patient to music. Eligible patients were approached and written informed consent by patient or legal representative was obtained.

### Intervention
All participating subjects were treated according to standard of care. Method of music intervention administration (ie, type, frequency and duration) was applied based on previous studies.[17 18] Participants in the intervention group (ie, music group) received over-ear headphones and a tablet with access to a platform with different preselected music playlists (ie, jazz, blues, classical, electronic, pop, 60s, 70s, 80s, etc), in which the music selection could

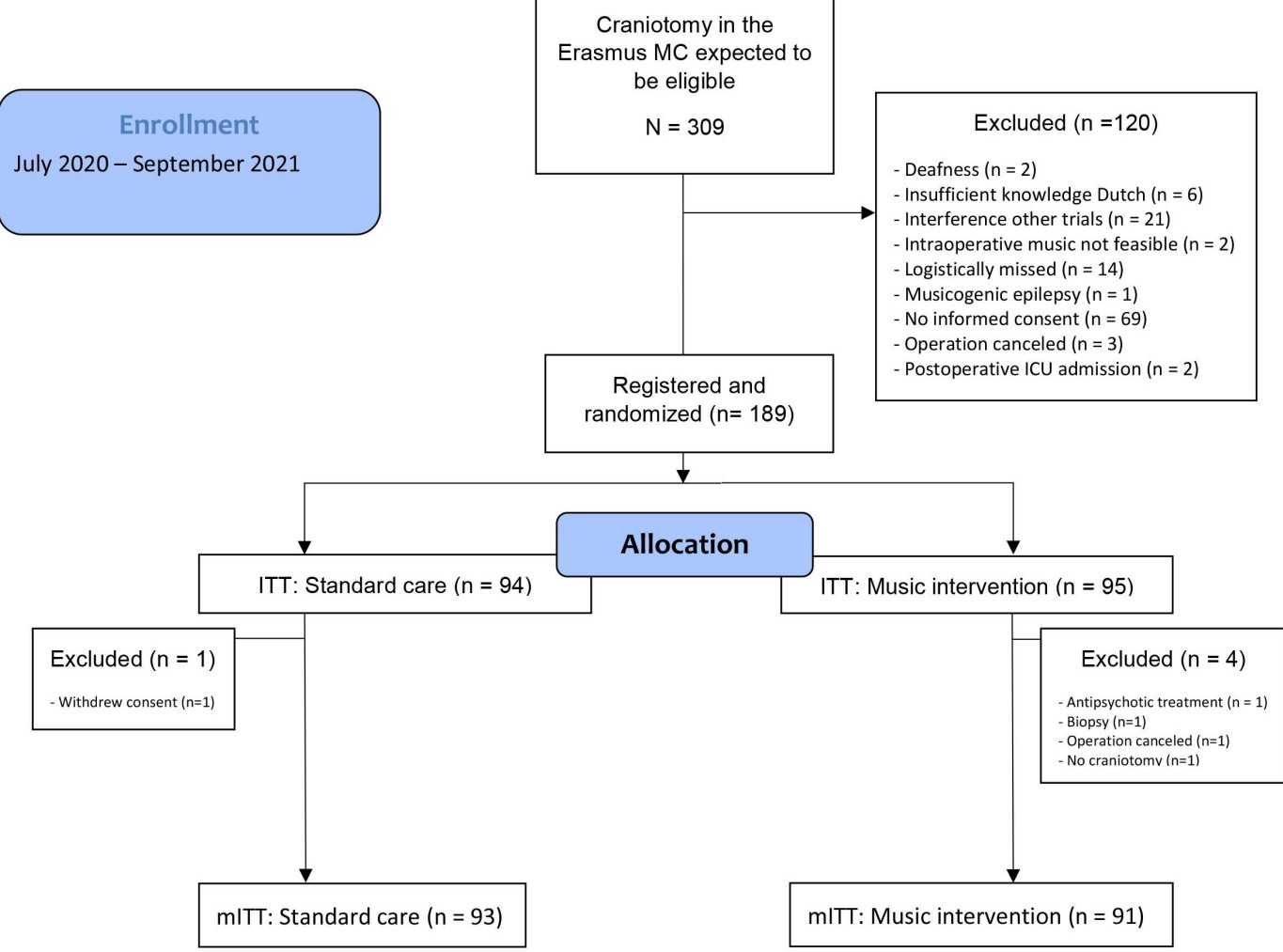

**Figure 1** CONSORT flow chart. CONSORT, Consolidated Standards of Reporting Trials; ICU, intensive care unit; ITT, intention-to-treat; MC, Medical Center; mITT, modified ITT.

be extended based on patients' wishes. These patients received the first 30 min of music at the preoperative holding area the day of operation (day 0) while awaiting surgery (see online supplemental figure 1 for 'Study Course MUSYC trial') which was stopped before reaching the operating room. In the operating room, in-ear earphones after intubation, which were compatible with the Mayfield clamp and site of operation, were inserted and music was continued until just before detubation. After surgery, during recovery at the postoperative care unit, another 30 min of recorded music through over-ear headphones was administered. Finally, participants received 30 min of recorded music twice a day until postoperative day 3. Patients in the control group were asked to refrain from music listening; however, this was not strictly controlled as this would influence the standard of clinical care too much. Nurses were instructed to monitor for music listening behaviour with a diary which was placed next to the music equipment. Periodically (approximately every 6 months), training was given for all nurses on the ward to explain how music had to be administered and monitored.

### Outcome measures

The primary outcome measure was presence or absence of postoperative delirium within the first 5 postoperative days.[27] The diagnosis of delirium required a two-step procedure; first, all participating patients were daily screened by the treating nurse using the DOSS, a validated 13-item delirium screening tool with higher scores indicating a higher probability of delirium. Use of the DOSS was already current practice at our department and was administered by the nurse during each shift (three 8-hour shifts per day).[28–31] Second, in case of a daily mean score of 3 or higher, which was radiologically not substantiated by a neurosurgical complication, a psychiatrist was consulted to assess the clinical diagnosis of delirium based on the DSM-5 criteria.[32] DSM-5 criteria assessment was conducted once in case of an increased mean DOSS score of 3 or above. This was not standardised for a certain moment of the day, but depended on the timing of the increased mean DOSS score and the logistic of the consulting psychiatrist that specific day. We chose not to blind the assessors from the intervention, as this could not be secured, which might have led to misleading results.

**Table 1** Baseline characteristics

|  | Control (n=94) | Music (n=95) |
|---|---|---|
| Prehospital demographics |  |  |
| Age (years)* | 61 (51–69) | 60 (49–69) |
| Sex (% female) | 46 (49) | 38 (40) |
| Medical history, n (%) |  |  |
| Somatic history† | 79 (84) | 74 (78) |
| Psychiatric history | 14 (15) | 6 (6) |
| Delirium prior admission | 2 (2) | 5 (3) |
| Dementia | 0 (0) | 0 (0) |
| Body mass index (kg/m$^2$) | 26 (24–30) | 26 (23–28) |
| Medication, n (%)‡ | 15 (26) | 14 (15) |
| Intoxication§ |  |  |
| Abuse of alcohol, n (%) | 6 (6) | 1 (1) |
| Abuse of drugs, n (%) | 3 (3) | 3 (3) |
| In-hospital demographics |  |  |
| Prehospital functional status¶ | 80 (70–90) | 80 (70–90) |
| KPS (100–0) | 70 (18–80) | 70 (45–80) |
| MRS (5–1) | 1.0 (0.25–2.0) | 1.0 (1.0–2.0) |
| Quality of life (1–100) | 70 (60–79) | 70 (55–80) |
| Cognitive function (0–30)** | 24 (19–27) | 25 (21–27) |
| Electrolyte disturbance, n (%)†† | 10 (11) | 9 (10) |
| Disease characteristics |  |  |
| Neurological deficit, n (%) | 35 (37) | 36 (38) |
| Type |  |  |
| Oncological | 81 (86) | 80 (84) |
| Vascular | 12 (13) | 14 (15) |
| Other | 1 (1) | 1 (1) |
| Frontal disease localisation, n (%) | 40 (43) | 24 (25) |
| Music affection |  |  |
| Music importance | 7.0 (6.8–8.0) | 8.0 (7.0–8.0) |
| Frequency listening |  |  |
| The whole day | 29 (31%) | 21 (22%) |
| Some hours per day | 44 (47%) | 54 (57%) |
| Some hours per week | 9 (10%) | 12 (13%) |
| Never | 6 (6%) | 3 (3%) |
| Played an instrument | 20 (21%) | 20 (21%) |
| Operation details‡‡ |  |  |
| Operation duration (min) | 220 (160–320) | 210 (140–290) |

Continued

**Table 1** Continued

|  | Control (n=94) | Music (n=95) |
|---|---|---|
| Emergency operation, n (%)§§ | 2 (2) | 4 (4) |
| Supine position, n (%) | 81 (86) | 80 (84) |
| Tramrail sign tension, n (%) | 68 (72) | 67 (7) |

*All continuous data are presented in median/IQR.
†Somatic history: including systematic disease (DM, hypertension) currently treated by medication and prior surgery (requiring general anaesthesia on the operating room).
‡Medication known to induce delirium before admission, such as sleep medication, morphine, atropine and antidepressants.
§Reported abusive use of alcohol and/or drugs.
¶Patients' functional performance with the KPS (ranging from 100/'no complaints' to 0/'death') and MRS (ranging from 0/'no symptoms to 5/'death'.
**Cognitive function assessed with the Montreal Cognitive Assessment.
††Electrolyte disturbance (mEq/L) in case of sodium >145 or <135 or potassium <3.5 or >5.
‡‡Patients in the mITT population (n=184) only.
§§Operation indication within 72 hours.
DM, diabetes mellitus; KPS, Karnofsky Performance Scale; mITT, modified intention-to-treat; MRS, Modified Rankin Scale.

Secondary outcomes were assessed to substantiate the effects of music on delirium and its clinical implications. Preoperative secondary outcomes (during the 30-minute preoperative holding stay) included anxiety (measured with the Visual Analogue Scale-anxiety/VAS-A) and heart rate variability (HRV), a marker of the autonomic tone reflecting parasympathetic nervous activity, measured with a 30-minute ECG recording. The following HRV parameters were analysed: SD of normal sinus beats, root mean square of successive differences between normal heartbeats (RMSSD), the number of adjacent NN intervals that differ from each other by more than 50 ms and the ratio of low frequency to high frequency power. Intraoperative secondary outcomes included depth of anaesthesia with Bispectral Index (BIS, Aspect V.3.22) with standardised sedation dosages (propofol and remifentanil). BIS was measured from the non-operated side, if feasible with site of resection, and the anaesthesiologist was blinded from the intraoperative BIS values, which was considered ethical as this form of monitoring is not standard of clinical care during intracranial procedures. Postoperative secondary outcomes (measured during the entire postoperative stay) were delirium severity (using the Delirium Rating Scale-revised-98) and delirium duration (onset until first day DOSS score <3), pain (Numerical Rating Scale pain and dosages of analgesic), postoperative complications, length of stay and location of discharge. Finally, patients' satisfaction of receiving

**Table 2** Primary outcome

| | Control (n/%) | Music (n/%) | Univariable analysis (OR/95% CI) | P value* | Multivariable analysis (OR/95% CI) | P value† |
|---|---|---|---|---|---|---|
| Intention-to-treat analysis (ITT) | | | | | | |
| Increased DOSS | 21/22.3 | 11/11.6 | 0.46/0.19, 1.00 | 0.048 | 0.49/0.20, 1.00 | 0.050 |
| Confirmed by DSM-5 | 7/7.4 | 4/4.2 | 0.55/0.14, 1.96 | 0.342 | 0.57/0.16, 2.07 | 0.39 |
| Modified ITT | | | | | | |
| Increased DOSS | 21/22.6 | 11/12.1 | 0.47/0.21, 1.04 | 0.060 | 0.47/0.16, 2.07 | 0.064 |
| Confirmed by DSM-5 | 7/7.5 | 4/4.4 | 0.57/0.14, 2.03 | 0.370 | 0.58/0.16, 2.10 | 0.412 |
| Safety population | | | | | | |
| Increased DOSS | 21/22.3 | 11/12.0 | 0.47/0.21, 1.04 | 0.061 | 0.47/0.21, 1.04 | 0.064 |
| Confirmed by DSM-5 | 7/7.4 | 4/4.3 | 0.58/0.14, 2.03 | 0.370 | 0.58/0.16, 2.11 | 0.414 |

*P values assessed with the $X^2$ test.
†Logistic regression analysis with groups, type disease and gender as independent variables.
DOSS, Delirium Observation Screening Scale; DSM-5, latest Diagnostic and Statistical Manual of Mental Disorders.

music was assessed with a 100-point VAS (administered at the outpatient clinic 6 weeks after discharge).

Baseline characteristics were extracted at baseline from questionnaires or the electronic patient file consisting of age, gender, medical history, daily function (Karnofsky Performance Scale (KPS) or Modified Rankin Scale (MRS)), quality of life (100-Likert scale, EuroQol(EQ)-5D and European Organization for Research and Treatment of Cancer (EORTC) -BN20), cognitive function (Montreal Cognitive Assessment (MoCA)), disease characteristics (ie, neurological deficit, type and side of intracranial pathology) and operation details (ie, emergency grade, duration of surgery).

## Statistical analysis

The main analysis was the comparison of the proportion of patients with delirium between the two arms in the intention-to-treat population (ITT; all registered and randomised patients) using univariate logistic regression. As sensitivity analyses, the proportion of patients with delirium was also compared between the two arms in the modified ITT (mITT, that is, ITT but excluding patients who were found to be ineligible after randomisation) and safety population (ie, all patients who underwent craniotomy). A multivariable logistic regression analysis with the stratification factors in the ITT population was also performed as sensitivity analysis, while multivariable analyses in the mITT and safety population should be considered as descriptive and therefore as hypothesis generating only. All secondary outcomes were analyses in the mITT population and should only be considered as descriptive only.

A 2-hour recording of BIS (blinded from the anaesthesiologist between operation minutes 60 and 180) was split into samples of 15 min as time points. A 30-minute recording of HRV was split into samples of 5 min as time points. Subsequently, we ran a linear mixed model with unstructured covariance for BIS and HRV, as a within-subject variability was suspected, with time point and

interaction group/time point as independent variables, as presented with fixed effects (beta/$\beta_1$) and 95% CI. Moreover, a sensitivity analysis was conducted with possible additional confounding for BIS level including age, comorbidity, type of disease, American Society of Anesthesiologists classification and steroid use. The residual plots were visually observed and a log transformation was applied in case of heteroscedasticity.

A two-sided p value of 0.05 or less was considered statistically significant. All statistical analyses were conducted using R (V.4.1.1).

## RESULTS

A total of 309 patients were expected to be eligible after screening, of which 189 patients were registered and randomly assigned to the music (n=95) or control group (n=94). Five patients (four in the music group and one in the control group) were excluded after registration due to withdrawing consent (n=1), preoperative use of antipsychotic treatment (n=1) or no craniotomy (n=3, one operation cancelled, one burr-hole biopsy and one no necessity of opening the dura). The remaining 184 patients, constituting the mITT population, were followed up for all the secondary outcomes.

The baseline characteristics were similar in the two trial groups (ITT population), with a median age of 60 years and 44% being female (table 1). Psychiatric medical history was reported in 11%, including depression (n=10) in most cases, preoperative usage of possible delirium-eliciting medication (ie, antidepressants and sleep medication) in 15% and no dementia in our cohort. Baseline cognition (MoCA) was 24/20–27 (median/IQR), quality of life was 70/55–80 (median/IQR) and no neurological symptoms (in case of a KPS=100 or MRS=0) at admission were present in 23%. This cohort included mostly neuro-oncological patients (n=161, 85%), with neurological deficit present in 38% and frontal localisation in 34%.

**Table 3** Secondary outcomes

| | Control (n=93) | Music (n=91) | P value |
|---|---|---|---|
| **Univariable analyses** | | | |
| Anxiety difference (mean/SD)* | 0.05/0.94 | −0.25/1.49 | 0.058 |
| Pain (mean/SD)† | 3.56/1.91 | 3.16/1.74 | 0.246 |
| Naproxen mg (mean/SD) | 13.6/75.4 | 2.75/26.2 | 0.103 |
| Oxycodon mg (mean/SD) | 2.03/4.35 | 1.61/3.31 | 0.828 |
| No complications, n (%) | 25 (26.9) | 21 (23.1) | 0.551 |
| Length of stay, days (mean/SD) | 7.43 (8.08) | 6.74 (8.26) | 0.947 |
| Discharge home, n (%) | 77 (82.8) | 76 (83.5) | 0.896 |
| **Multivariable analyses** | | | |
| **Heart rate variability/HRV ($\beta_1$/95% CI)‡** | | | |
| Time point§ | RMSSD | NN50 | LF/HF |
| 5 min | 55.08/13.16, 97.00* | 17.64/−3.92, 39.21 | -0.46/−1.04, 0.11 |
| 10 min | 18.79/−16.16, 53.75 | 11.11/−8.67, 30.88 | 0.49/−0.18, 1.17 |
| 15 min | −3.20/−38.60, 32.19 | 9.14/−11.11, 29.38 | 0.34/−0.28, 0.96 |
| 20 min | 3.12/−40.54, 46.78 | 8.22/−12.00, 28.45 | 0.56/−0.07, 1.19 |
| 25 min | −11.85/−58.03, 34.33 | −3.56/−27.10, 19.98 | 0.45/−0.26, 1.16 |
| 30 min | 22.20/−17.01, 61.41 | −5.98/−28.49, 16.53 | 0.46/−0.18, 1.09 |

| Time point§ | Depth of anaesthesia/BIS ($\beta_1$/95% CI)¶ | | P value |
|---|---|---|---|
| 15 min | 0.71/−3.17, 4.58 | | 0.717 |
| 30 min | −1.44/−3.34, 0.46 | | 0.139 |
| 45 min | −1.06/−3.42, 1.30 | | 0.378 |
| 60 min | −2.23/−5.34, 0.89 | | 0.162 |
| 75 min | −1.82/−5.48, 1.83 | | 0.328 |
| 90 min | −2.46/−6.72, 1.79 | | 0.256 |
| 105 min | −0.50/−6.17, 5.16 | | 0.862 |
| 120 min | 0.31/−5.44, 6.05 | | 0.917 |

Secondary outcomes analysed on the mITT population.
*Anxiety differences between first and second measures with VAS-A.
†Pain (NRS) over the first 3 postoperative days.
‡HRV analyses: 30 min of preoperative ECG recordings split into 5-minute samples; all values marked with * are significant (ie, p<0.05).
§Time point samples included in the linear mixed model.
¶BIS analyses: 120 min of intraoperative BIS registration split into 15-minute samples.
BIS, Bispectral Index; LF/HF, ratio of low frequency to high frequency power; mITT, modified intention-to-treat; NN50, number of adjacent NN intervals that differ from each other by more than 50 ms; NRS, Numerical Rating Scale; RMSSD, root mean square of successive differences between normal heartbeats; VAS-A, Visual Analogue Scale-anxiety.

Affection for music was reported with an importance of 8/7–8 (median/IQR); only 5% reported never to listen music in daily life. Surgical details showed duration of surgery of 220 min and emergency surgery (ie, within 72 hours) in only 3%.

## Primary outcome

In the music group, adherence to the music intervention before, during, and directly after surgery was 96%, 100%, and 74%, respectively (online supplemental table 1). The following days, the adherence decreased each day, from 70% on the first morning to 47% at noon on day 3. The total listening time was a median of 130 min (IQR, 73–230) during the 5 days of admission or until discharge.

A high DOSS score (ie, 3 or higher) was observed in 32 patients. This was caused by a neurosurgical complication, as confirmed on radiology, in three patients: two patients with infarction after a vascular procedure with hemiparesis and decreased attention. The other patient had a subdural haematoma which was evacuated in the operating room. This resulted in 29 patients with possible delirium by DOSS; 21 were evaluated by the psychiatrist who diagnosed delirium in 11 patients (57%, online supplemental figure 2) based on the DSM-5 criteria.

According to the DOSS, a significantly higher incidence of delirium was observed in the control (n=21) versus music (n=11) group in the ITT population for the univariable (22.3% vs 11.6%, p=0.048) and multivariable

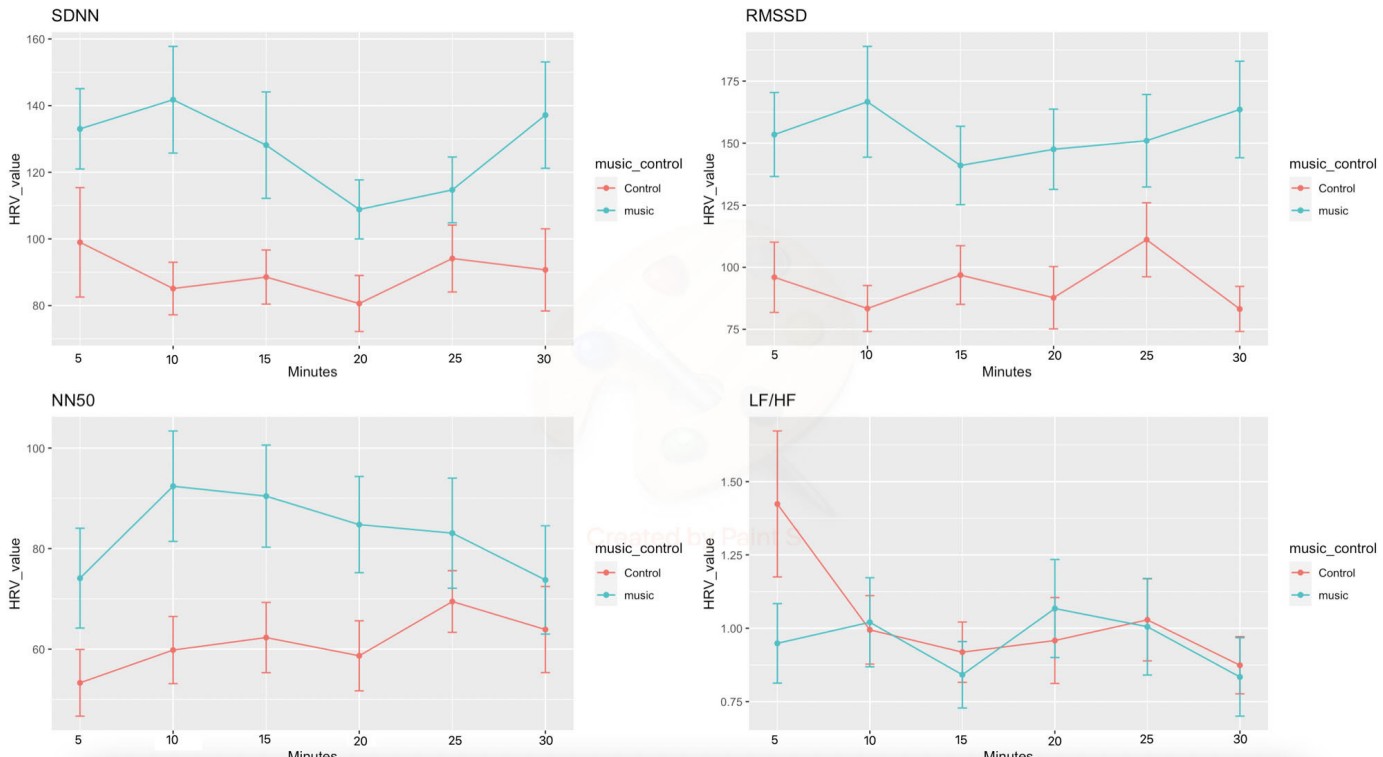

**Figure 2** Preoperative course of HRV. A 30-minute recording of HRV was split into samples of 5 min as time points and compared between groups. Heart rate remained constant in the music group while it decreased after 15 ($\beta_1$=2.89, p=0.043), 25 ($\beta_1$=3.36, p=0.05) and 30 ($\beta_1$=5.06, p=0.011) min in the control group. A significant increase on HRV was found by music at 5 min on RMSSD ($\beta_1$=55.08, p=0.012). No significant effect was found on the other HRV parameters. HRV, heart rate variability; LF/HF, ratio of low frequency to high frequency power; NN50, number of adjacent NN intervals that differ from each other by more than 50 ms; RMSSD, root mean square of successive differences between normal heartbeats; SDNN, SD of normal sinus beats.

(OR/95% CI: 0.49/0.20 to 1.00, p=0.050) analyses. This was not observed in the mITT (p=0.064) and the safety (p=0.064) population. The occurrence of a DSM-5-confirmed delirium was not statistically significant between the control (n=7) versus music (n=4) group in the ITT (7.4% vs 4.2%, OR=0.55), the mITT (7.5% vs 4.4%, OR=0.57) and the safety (7.4% vs 4.3%, OR=0.58) population (table 2).

Of those patients with DSM-5-confirmed delirium (n=11), severity of delirium (mean/SD) was 12.60/5.52, which was not different between the two arms (p=0.857). The duration of delirium (days, mean/SD) was 3.36/4.69, which was not different between the two arms (p=0.761).

### Secondary outcome

Available ECG data (n=87) revealed that heart rate remained constant in the music group while it decreased after 15 ($\beta_1$=2.89, p=0.043), 25 ($\beta_1$=3.36, p=0.05) and 30 ($\beta_1$=5.06, p=0.011) min in the control group (table 3 and figure 2). A significant increase on HRV was found by music at 5 min on RMSSD ($\beta_1$=55.08, p=0.012). No significant effect was found on the other HRV parameters. Available depth of anaesthesia (n=70) data revealed no significant difference between the music and control group at the several analysed time points (figure 3). A trend towards less anxiety in the music group was observed

(p=0.058, figure 4). All other secondary outcomes were not different between groups.

Patient satisfaction (median/IQR) in the music group who filled in the questionnaire (n=68) was 85/80–95, and 88% reported they would want to receive music in case of future surgery.

### DISCUSSION

We found a significant decrease on the incidence of postoperative delirium by the addition of music perioperatively using the DOSS; however, this was not significant when assessed by the DSM-5 criteria. Second, music activated preoperative HRV, a marker of autonomic tone. Last, no significant effects on anxiety, depth of anaesthesia, postoperative complications, length of stay and location of discharge were found. Clinical implications and limitations are discussed below.

We found a significant decrease on the incidence of postoperative delirium, defined with DOSS, by the addition of perioperative music. A recent published systematic review conducted a meta-analysis on the preventive effect of music on delirium with six studies and found a relative reduction similar to ours (0.52 vs 0.48).[22] This meta-analysis included pilot studies which administered

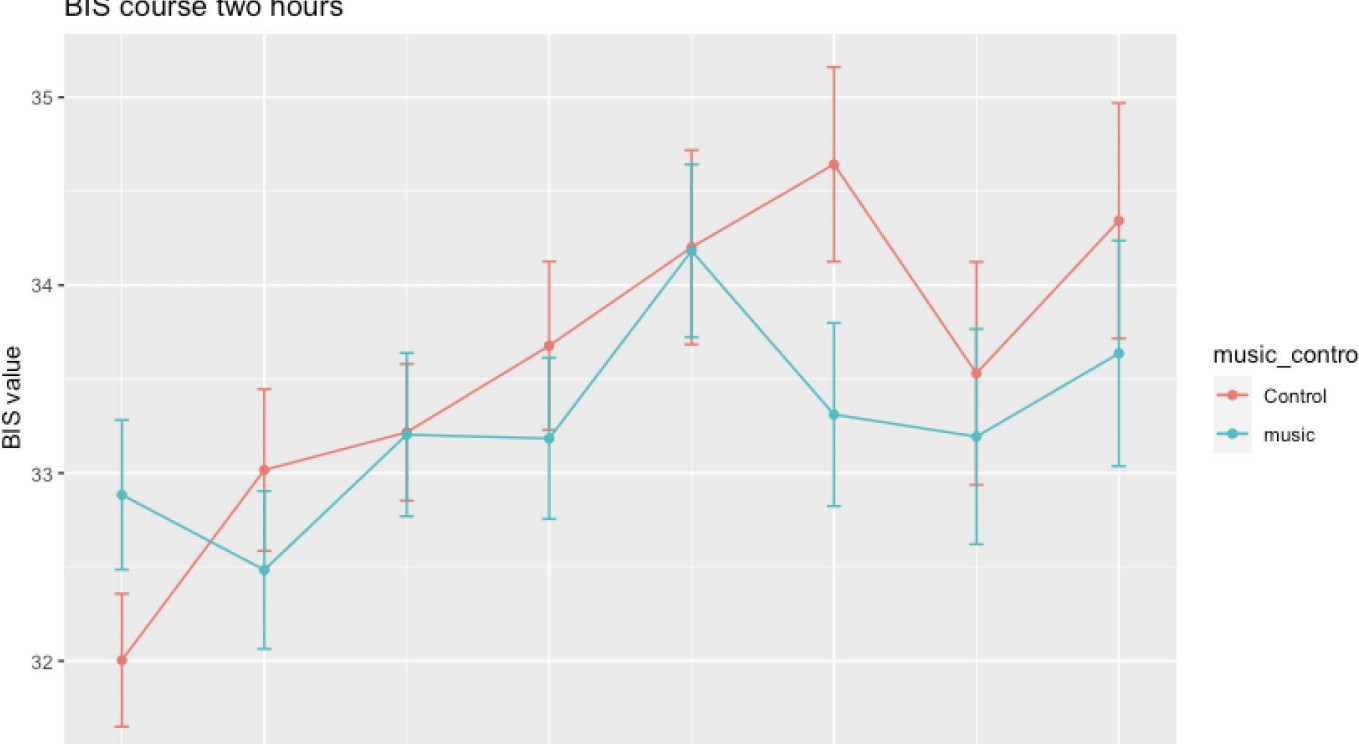

**Figure 3** Intraoperative depth of anaesthesia. A 2-hour recording of BIS was split into samples of 15 min as time points and compared between groups. Available depth of anaesthesia (n=70) data revealed no significant difference between the music and control group at the several analysed time points. BIS, Bispectral Index.

patient-preferred music similar to our design, but varied from ours as the population considered surgical and non-surgical ICU patients and delirium was defined with the Neelon and Champagne Confusion Scale (NEECHAM), Confusion Assessment Method-ICU or own definitions. Our sample size calculation was based on other neuro-surgical studies evaluating delirium in case of increased scores on delirium screening tools.[1] When handling screening tools for delirium definition by using the DOSS, we support the efficacy of music in lowering the incidence of postoperative delirium. However, although a similar trend was found, significance of results was not achieved when assessed by the DSM-5 criteria. The discrepancy between DOSS and DSM-5 may have several explanations. First, DSM-5 was evaluated by a psychiatrist *after* an increased DOSS score. Hence, delirium may have been resolved over time before the psychiatrist's assessment. Moreover, DOSS evaluation was conducted three times per day by the nurses, as opposed to DSM-5 determination which was only evaluated once. DSM-5 assessment during daytime might have missed some cases as delirium fluctuates over the course of the day, especially for the delirium type present during night-time. Also, not all our patients with increased DOSS were evaluated by a psychiatrist due to logistics and we might have missed some patients with delirium. Second, delirium screening tools have not been validated within the neurosurgical

population.[5–8 33–37] Hence, while high diagnostic accuracies in the general population justify diagnostic usage of delirium screening tools, it is unclear whether this can be adopted to our complicated patient population, as a positive screen for delirium may be due to the underlying neurological disease or its sequelae (eg, oedema, vasospasm, seizures, rebleeding, ischaemia) leading to false-positive results.

We propose a vagal-mediated anti-inflammatory response as a candidate pathway of music on delirium, as hypotheses of delirium rely on neuroinflammatory reactions within the brain.[2] Although we did not assess inflammatory cytokines in our study, vagal nerve activation by music was supported by the increased HRV, revealed by an increased RMSSD on ECG during the preoperative music session. The activation of HRV by music in brain-damaged patients was proven earlier and is considered a valid marker of parasympathetic nervous activation.[38 39] Whether HRV could be used as a marker for postoperative recovery remains to be determined. Moreover, we observed a decreasing trend in preoperative anxiety by music, although this did not reach significance. This preoperative parasympathetic activation and anxiety reduction may have induced a sedative-sparing effect, subsequently increasing cortical engagement and cognitive processing.[40] We did not find a deeper level of anaesthesia in the music group. Literature is contradictory

Pre-operative course anxiety

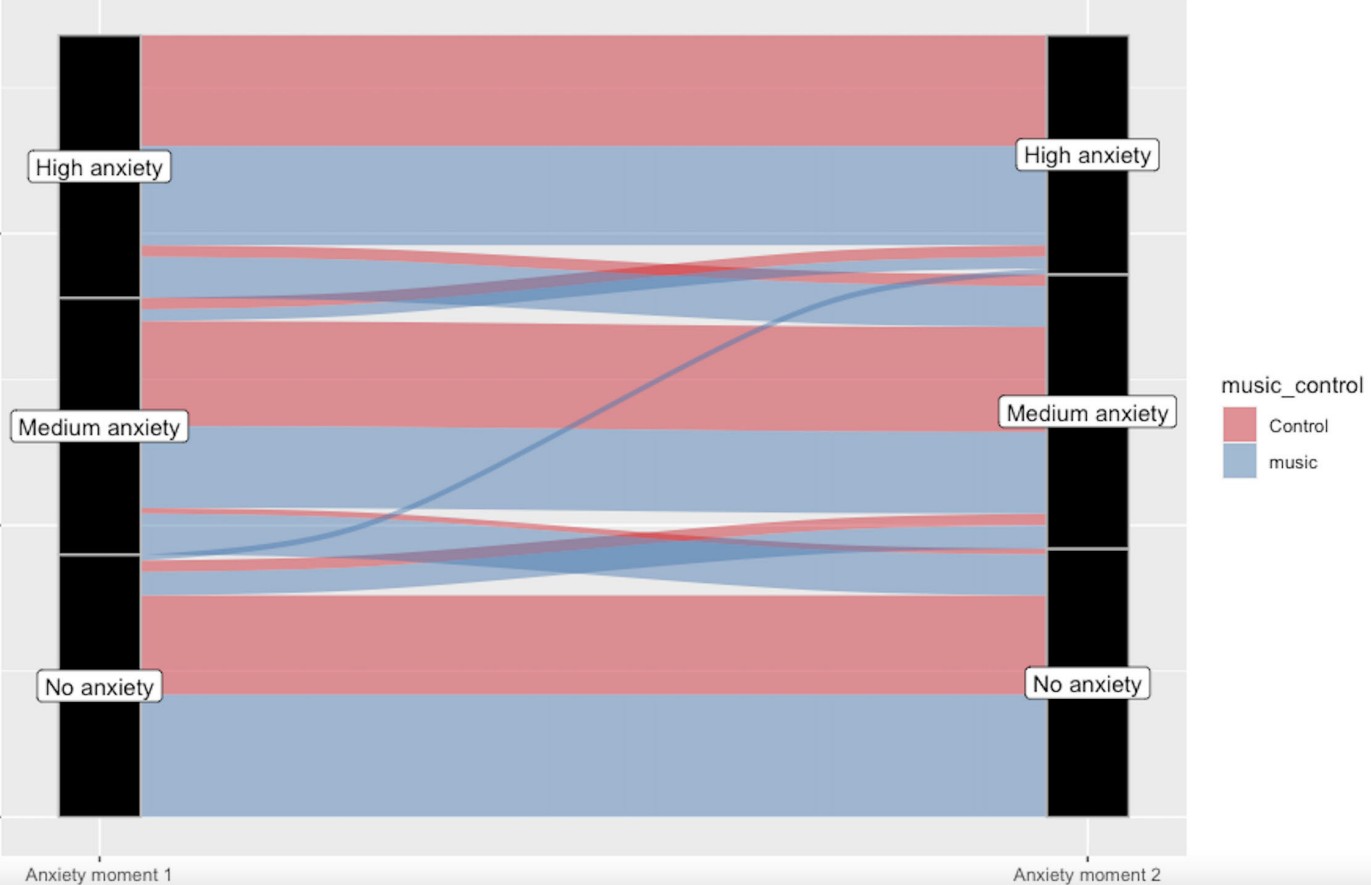

**Figure 4** Two anxiety measurements (VAS-A) were conducted before surgery and categorised in 'no anxiety' (0–2), 'medium anxiety' (3–6) and 'high anxiety' (7–10). Most patients remained in their anxiety level, although some patients showed a decreased preoperative anxiety when receiving music, but this difference was non-significant between groups (p=0.058). VAS-A, Visual Analogue Scale-anxiety.

on the correlation of depth of sedation and anaesthetic requirements as music listening is associated with decrease of depth of anaesthesia, but no decrease of sevoflurane was achieved when pursuing constant depth of anaesthesia. Future neurosurgical studies should confirm whether concentration of sedation can be reduced with music in case of standardised depth of anaesthesia (BIS) levels.

We found high adherence to the music intervention before surgery. High importance of music in daily life, the number of hours listening to music in daily life and the willingness to receive music intervention in case of future surgeries were found in our cohort, which are considered important facilitators for music implementation.[41] However, the adherence declined after surgery, due to pain, nausea, logistics (ie, for MRI) or unwillingness. Absence of a near future operation prospect may have reduced the urgent necessity of music, resulting in the postoperative decline in adherence. Lack of knowledge of the intervention is considered a barrier for implementation. Informing patients, substantiated by the results from efficacy studies such as this trial, may aid in the implementation of music in the neurosurgical population.

Although delirium is (most often) temporary and self-limiting, delirium independently predicts clinically relevant outcomes in neurologically damaged patients.[9] [10] Although a trend was observed, we did not find any significant positive effects on complications, length of admission or location of discharge. Future studies should assess the long-term implications of delirium defined with either DOSS and DSM criteria after discharge in neurosurgical patients.

### Strengths and limitations

This is the first randomised controlled trial assessing the effect of music on delirium after craniotomy and the largest assessing the effect of music against delirium. We showed that music reduced the incidence of delirium when defined with the DOSS but not after DSM-5 confirmation. Our study was subject to several limitations: first, not all our patients with increased DOSS were evaluated by a psychiatrist due to logistics and we might have missed some patients with delirium. However, we feel that this did not affect our conclusions, because, with our low confirmation rate of suspected delirium, the study would still have been underpowered. Second, due to the nature of

the intervention, we did not blind the study, which could have influenced the outcome assessors. However, blinding could not be secured which might have led to misleading results. Third, the generalisability of the results may be affected by the single-centre design of the study.

## Conclusion

Our results support the efficacy of music in reducing the incidence of delirium after craniotomy, as found with DOSS but not after DSM-5 confirmation. Delirium screening tools should be validated within the neuro-surgical context and the long-term implications of a delirium, either defined by an increased DOSS or DSM-5, should be evaluated. This effect of music is substantiated by the effect of music on an increased preoperative HRV. Last, although preoperative adherence was high, this declined after surgery, which should be taken into account when considering implementation in the neuro-surgical population.

**Author affiliations**
[1]Department of Neurosurgery, Erasmus Medical Center, Rotterdam, The Netherlands
[2]Department of Psychiatry, Erasmus Medical Center, Rotterdam, The Netherlands
[3]Institute for Medical Technology Assessment (iMTA), Erasmus Medical Center, Rotterdam, The Netherlands
[4]Department of Paediatric Surgery and Intensive Care, Erasmus Medical Center, Rotterdam, The Netherlands
[5]Department of Cardiology, Erasmus Medical Center, Rotterdam, The Netherlands
[6]Department of Haematology, Erasmus Medical Center, Rotterdam, The Netherlands
[7]Department of Anesthesiology, Erasmus Medical Center, Rotterdam, The Netherlands

**Contributors** Each author has contributed significantly to, and is willing to take public responsibility for, one or more aspects of the study. AJPEV, CMFD, JJ, MK and PRK conceived the study idea and are guarantor of the project. PRK coordinated the research protocol. PRK and MIM extracted the data. PRK, BvdH and AJPEV conducted the statistical analysis and wrote the first draft of the manuscript. JJ, MIM, CMFD, MK, SK, R-JO, MC, MJP, MvS and AJPEV critically revised the manuscript. All authors have seen and approved the final version of the manuscript being submitted. The article is the authors' original work, has not received prior publication and is not under consideration for publication elsewhere.

**Funding** This research is funded with the Erasmus Medical Center Efficiency Grant (grant number: 19105) by the Erasmus Medical Center, Rotterdam, the Netherlands.

**Competing interests** None declared.

**Patient and public involvement** Patients and/or the public were involved in the design, or conduct, or reporting, or dissemination plans of this research. Refer to the Methods section for further details.

**Patient consent for publication** Not required.

**Ethics approval** This study involves human participants. The Medical Ethics Review Committee of the Erasmus Medical Center, Rotterdam (Erasmus MC), which declared this study not subject to the Medical Research Involving Human Subjects Act (ie, 'non-WMO'), approved this study (MEC-2020-0064) on 25 February 2020, as such. Written informed consent was obtained for each inclusion by patient or legal representative.

**Provenance and peer review** Not commissioned; externally peer reviewed.

**Data availability statement** Data are available upon reasonable request. The raw data supporting the conclusions of this article will be made available by the authors, without undue reservation.

**ORCID iDs**
Pablo R Kappen http://orcid.org/0000-0003-4987-297X
Mathijs S van Schie http://orcid.org/0000-0001-6695-5465
M Klimek http://orcid.org/0000-0002-0122-9929

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
