## [Reviewer comments · BMJ Open]

ARTICLE DETAILS

TITLE (PROVISIONAL)	Music to prevent delirium during neurosurgery (MUSYC): a single-centre prospective randomised controlled trial.
AUTHORS	Kappen, Pablo; Mos, M.I.; Jeekel, Johannes; Dirven, Clemens; Kushner, Steven; Osse, Robert-Jan; Coesmans, Michiel; Poley, Marten; van Schie, Mathijs; van der Holt, Bronno; Klimek, M; Vincent, Arnaud

VERSION 1 – REVIEW

REVIEWER	Litofsky , Scott University of Missouri, Division of Neurological Surgery , University of Missouri School of Medicine , Columbia , MO , USA.
REVIEW RETURNED	11-Dec-2022

GENERAL COMMENTS	The authors have evaluated the effects of music therapy on post-operative delirium by comparing a group of 91 patients treated with music to 93 patients who did not receivedd music therapy. They a found a higher incidence of delirium in the control group (22.3%) compared to the music group (11.6%) using the DOSS scale. This result, however, was not confirmed by a psychiatric evaluation using DSM-5. The authors do not satisfactorily discuss why this difference in assessment method would be associated with different results. The authors also found that pre-operative heart rate decreased over time in the music group, but not the control group. The last major finding was that increased depth of anesthesia was observed in the music group relative to the control group. Unfortunately, logistic regression for confounders, such as age, benzodiazepine use, steroid use, ASA grade, comorbidities, and disease process was not performed. The manuscript is also significantly under-referenced, particular with regard to delirium in neurosurgical populations. The manuscript could be improved with attention to the following points: 1. In the ABSTRACT CONCLUSION, the authors should indicate that the incidence of delirium is reduced with music therapy, not that delirium is prevented. That same change is necessary in the CONCLUSIONS of the text, as well as throughout the manuscript.2. In the INTRODUCTION (line 20, page 6), the authors should state that delirium is neurosurgical populations also multi-factorial. Expanding the references here would be helpful.3. In INTRODUCTION, some background of DOSS and DSM-5 would be helpful.4. In METHODS Page 7, line 50), the authors should explain what is meant by "undergoing surgery impeding supply of music." Do they mean that headphones could not be placed, as with vestibular schwannoma surgery, for example?
---

	5. On page 7, line 60, is "interfering" the desired word, instead of "inferring." 6. In METHODS, when was music started and stopped prior to surgery? 7. In METHODS, what time of day was DOSS and DSM5 measured? 8. In METHODS, when were secondary outcomes measured? 9. In METHODS, logistic regression for confounders is necessary, as stated above. 10. In DISCUSSION (page 15, line 11 and elsewhere), the authors should make clear that the reduction in delirium is in "incidence." 11. Redundancy should be reduced, as in the first two paragraphs of DISCUSSION.
--	--

REVIEWER	Kose, Gulsah Mugla Sitki Kocman Universitesi
REVIEW RETURNED	19-Dec-2022

GENERAL COMMENTS	Congratulations on your research on an unresolved issue. My suggestions for your work are as follows Abstract section  • In the method section, information on how many patients were included in the study (study group and control group) and between which dates the study was carried out should be added. In the findings section, the information given for this should be removed. • It should be added how many patients were found to have delirium, and how many people were found to have delirium in the study and control groups. • Your said: "Our results support the efficacy of music in preventing delirium after craniotomy, as found with DOSS but not after DSM-5 confirmation, substantiated. ... And "Our results fit well within the current literature and support the implementation of music for the prevention of delirium also within the neurosurgical population." but since there is no validation with DSM-5, such a definitive claim would be wrong. You have given results on delirium in neurosurgery, but you have cited only one source, which is systematic review. There is a lot of work done on this subject. therefore, these studies should also be mentioned. Also, neurosurgery patients have different results even in spinal and cranial patients. Which rate is yours? The information on page 11, lines 2-17 should be deleted from here and moved to the study design section. The information on page 13, lines 3-6 should be deleted from here and moved to the study design section. At which stage of the research was the data on the basic characteristics of the patients collected? What data were collected from basic characteristics? This information should be added to the collection of research data. The information on page 13, lines 6-11 should be deleted from here and moved to the intervention section. Doesn't including patients with a psychiatric history and taking medication affect the results? Why were these patients included in the study? How were data on quality of life collected? Information on this should be added to the application section of the research. How was the study protocol applied in patients undergoing emergency surgery? Was there enough time? Why was no further analysis performed with patients who used alcohol and had a psychiatric history? Since these are important
---

	factors affecting delirium, it is necessary to examine whether they contributed to the development of delirium. References should be written in a certain writing format, all references should be re-examined in terms of spelling rules. References before 2010 should be removed from the references list
--	--

REVIEWER	Maruo, Kazushi
REVIEW RETURNED	27-Jan-2023

GENERAL COMMENTS	The statistical part of this manuscript seems to be generally acceptable, but only the following point should be considered. For linear mixed-effects models, was heteroscedasticity between time points taken into account? When comparing between groups at each time point, the covariance structure should be unstructured or, if an otherwise simple structure was used, sandwich variance should be used.
--

VERSION 1 – AUTHOR RESPONSE

REVIEWER 1

Dr. Scott Litofsky , University of Missouri

Comments to the Author:

The authors have evaluated the effects of music therapy on post-operative delirium by comparing a group of 91 patients treated with music to 93 patients who did not received music therapy. They a found a higher incidence of delirium in the control group (22.3%) compared to the music group (11.6%) using the DOSS scale. This result, however, was not confirmed by a psychiatric evaluation using DSM-5. The authors do not satisfactorily discuss why this difference in assessment method would be associated with different results. The authors also found that pre-operative heart rate decreased over time in the music group, but not the control group. The last major finding was that increased depth of anesthesia was observed in the music group relative to the control group. Unfortunately, logistic regression for confounders, such as age, benzodiazepine use, steroid use, ASA grade, comorbidities, and disease process was not performed. The manuscript is also significantly under-referenced, particular with regard to delirium in neurosurgical populations.

We would like to thank reviewer dr. Scott Litofsky (reviewer 1) for critically revising our manuscript. We feel this input has significantly contributed to the quality of our paper. We have dedicated a larger part in the discussion section on the difference in outcome between the assessment methods, which now reads as follows (page 13):

'The discrepancy between DOSS and DSM-5 may have several explanations. First, DSM-5 was evaluated by a psychiatrist after an increased DOSS score. Hence, delirium may have been resolved over time before the psychiatrist its assessment. Moreover, DOSS evaluation was conducted three times per day by the nurses, as opposed to DSM-5 determination which was only evaluated once. DSM-5 assessment during day-time might have missed some cases as delirium fluctuates over the course of the day, especially for the delirium-type present during night-time. Also, not all our patients with increased DOSS were evaluated by a psychiatrist due to logistics and we might have missed some patients with delirium. Second, delirium screening tools have not been validated within the

neurosurgical population. (5-8, 33-37) Hence, while high diagnostic accuracies in the general population justify diagnostic usage of delirium screening tools, it is unclear whether this can be adopted to our complicated patient population, as a positive screen for delirium may be due to the underlying neurological disease or its sequelae (e.g., oedema, vasospasm, seizures, rebleeding, ischemia) leading to false-positive results.'

Moreover, we have added a sensitivity analysis with possible confounding factors including age, comorbidity, type of disease, American Society of Anesthesiologists (ASA) classification and steroid use on the depth of anesthesia (BIS) analysis. We have improved our statistical model on depth of anesthesia, which has resulted in non-significant results. Adding the suggested possibly confounding factors did not change our conclusion.

Last, as suggested by the reviewer, we have added more neurosurgical references in the entire manuscript.

All other comments have been addressed below point by point.

The manuscript could be improved with attention to the following points:

1. In the ABSTRACT CONCLUSION, the authors should indicate that the incidence of delirium is reduced with music therapy, not that delirium is prevented. That same change is necessary in the CONCLUSIONS of the text, as well as throughout the manuscript.

We have changed this, as suggested by the reviewer, in the abstract and throughout the entire manuscript.

2. In the INTRODUCTION (line 20, page 6), the authors should state that delirium is neurosurgical populations also multi-factorial. Expanding the references here would be helpful.

We have changed this as suggested.

3. In INTRODUCTION, some background of DOSS and DSM-5 would be helpful.

We have adjusted the introduction accordingly by describing the DSM-5 more thoroughly and explain how screening tools have been developed to aid in the recognition of delirium. We chose to elaborate more on the DOSS in the methods section.

4. In METHODS Page 7, line 50), the authors should explain what is meant by "undergoing surgery impeding supply of music." Do they mean that headphones could not be placed, as with vestibular schwannoma surgery, for example?

We have explained this by adding some examples as suggested.

5. On page 7, line 60, is "interfering" the desired word, instead of "inferring."

We have adjusted this as suggested.

6. In METHODS, when was music started and stopped prior to surgery?

This has been added in the Methods section which reads as follows (page 7 and 8): *'These patients received the first 30 minutes of music at the pre-operative holding area the day of operation (day 0) while awaiting surgery (see supplementary figure 1 for 'Study Course MUSYC trial') which was stopped before reaching the operation room.'*

7. In METHODS, what time of day was DOSS and DSM5 measured?

This has been added in the Methods section which reads as follows (page 8): *'Use of the DOSS was already current practice at our department and was administered by the nurse during each shift (three- 8-hour shifts per day). Second, in case of a daily mean score of 3 or higher, which was radiologically not substantiated by a neurosurgical complication, a psychiatrist was consulted to assess the clinical diagnosis of delirium based on the DSM-5 criteria. DSM assessment was conducted once in case of an increased mean DOS score of 3 or above. This was not standardized for a certain moment of the day, but depended on the timing of the increased mean DOSS score and the logistic of the consulting psychiatrist that specific day.'*

8. In METHODS, when were secondary outcomes measured?

This has been described in the methods sections which reads as follows (page 8 and 9): *'Pre-operative secondary outcomes (during the 30-minute pre-operative holding stay) included anxiety (measured with the Visual Analogue Scale-anxiety/VAS-A) and heart rate variability (HRV), a marker of the autonomic tone reflecting parasympathetic nervous activity, measured with a 30 minutes electrocardiography (ECG) recording. ... Intra-operative secondary outcomes included depth of anaesthesia with Bispectral Index (BIS, Aspect™ version 3.22) with standardized sedation dosages (propofol and remifentanil).... Post-operative secondary outcomes (measured during the entire post-*

operative stay) were delirium severity... discharge. Finally, demographic data were collected through a questionnaire at baseline and patients' satisfaction of receiving music was assessed with a 100- point visual analogue scale (administered at the outpatient clinic 6 weeks after discharge).'

9. In METHODS, logistic regression for confounders is necessary, as stated above.

A sensitivity analyses was conducted with possible additional confounding for depth of anaesthesia (BIS level) including age, comorbidity, type of disease, American Society of Anesthesiologists (ASA) classification and steroid use.

10. In DISCUSSION (page 15, line 11 and elsewhere), the authors should make clear that the reduction in delirium is in "incidence."

We have adjusted this in the discussion section as suggested by the reviewer.

11. Redundancy should be reduced, as in the first two paragraphs of DISCUSSION.

We have reduced redundancy throughout the entire manuscript as suggested by the reviewer.

REVIEWER 2

Dr. Gulsah Kose, Mugla Sitki Kocman Universitesi

Comments to the Author:

Dear author

Congratulations on your research on an unresolved issue. My suggestions for your work are as follows

We would like to thank reviewer dr. Gulsah Kose (reviewer 2) for critically revising our manuscript. We feel this input has significantly contributed to the quality of our paper. All the reviewers' comments were addressed below point by point.

Abstract section

- In the method section, information on how many patients were included in the study (study group and control group) and between which dates the study was carried out should be added. In the findings section, the information given for this should be removed.

We have adjusted this as suggested by the reviewer.

- It should be added how many patients were found to have delirium, and how many people were found to have delirium in the study and control groups.

We have added these numbers within the results section as suggested by the reviewer.

- Your said: “Our results support the efficacy of music in preventing delirium after craniotomy, as found with DOSS but not after DSM-5 confirmation, substantiated.... And “Our results fit well within the current literature and support the implementation of music for the prevention of delirium also within the neurosurgical population.” but since there is no validation with DSM-5, such a definitive claim would be wrong.

As stated in the discussion. Current neurosurgical literature defines delirium through increased scores on the delirium screening tools. When handling this definition, we support the efficacy of music in lowering the incidence of post-operative delirium. Although a similar trend was found, significance of results was not achieved when assessed by the DSM-5 criteria. This discrepancy between assessment methods deserves further explanation, which we elaborate upon and reads as follows (page 13): *Our sample size calculation was based on other neurosurgical studies evaluating delirium in case of increased scores on delirium screening tools. (2) When handling screening tools for delirium definition by using the DOSS, we support the efficacy of music in lowering the incidence of post-operative delirium. Although a similar trend was found, significance of results was not achieved when assessed by the DSM-5 criteria. The discrepancy between DOSS and DSM-5 may have several explanations. First, DSM-5 was evaluated by a psychiatrist after an increased DOSS score. Hence, delirium may have been resolved over time before the psychiatrist its assessment. Moreover, DOSS evaluation was conducted three times per day by the nurses, as opposed to DSM-5 determination which was only evaluated once. DSM-5 assessment during day-time might have missed some cases as delirium fluctuates over the course of the day, especially for the delirium-type present during night-time. Also, not all our patients with increased DOSS were evaluated by a psychiatrist due to logistics and we might have missed some patients with delirium. Second, delirium screening tools have not been validated within the neurosurgical population. (5-8, 33-37) Hence, while high diagnostic accuracies in the general population justify diagnostic usage of delirium screening tools, it is unclear whether this can be adopted to our complicated patient population, as a positive screen for delirium may be due to the underlying neurological disease or its sequelae (e.g., oedema, vasospasm, seizures, rebleeding, ischemia) leading to false-positive results.’*

You have given results on delirium in neurosurgery, but you have cited only one source, which is systematic review. There is a lot of work done on this subject. therefore, these studies should also be

mentioned. Also, neurosurgery patients have different results even in spinal and cranial patients. Which rate is yours???

We have added more references on neurosurgery and delirium within the manuscript. For the incidence we have used the pooled estimate of all the cranial surgery papers on delirium as described in our systematic review (Kappen PR, Kakar E, Dirven CMF, van der Jagt M, Klimek M, Osse RJ, et al. *Delirium in neurosurgery: a systematic review and meta-analysis. Neurosurg Rev. 2021.*) which was also in line with our retrospective cohort study (Kappen PR, Kappen HJ, Dirven CMF, Klimek M, Jeekel J, Andrinopoulou ER, et al. *Postoperative Delirium After Intracranial Surgery: A Retrospective Cohort Study. World Neurosurg. 2023.*) and concordant with this trial when handling the DOSS, but not the DSM.

The information on page 11, lines 2-17 should be deleted from here and moved to the study design section. The information on page 13, lines 3-6 should be deleted from here and moved to the study design section.

We have moved these lines from the results towards the methods section as suggested by the reviewer.

At which stage of the research was the data on the basic characteristics of the patients collected? What data were collected from basic characteristics? This information should be added to the collection of research data.

We have collected the demographic data at baseline and described this more thoroughly which now reads as follows (page 9): *'Baseline characteristics were extracted from questionnaires at baseline or the electronic patient file consisting of age, gender, medical history, daily function (Karnofsky Performance Scale/KPS or Modified Rankin scale/MRS), quality of life (100-likert scale, EQ-5D and EORTC QLQ – BN-20), cognitive function (Montreal Cognitive assessment / MoCA), disease characteristics (i.e. neurologic deficit, type and side of intracranial pathology) and operation details (i.e. emergency grade, duration of surgery).'*

The information on page 13, lines 6-11 should be deleted from here and moved to the intervention section.

We have moved these lines as suggested by the reviewer.

Doesn't including patients with a psychiatric history and taking medication affect the results? Why were these patients included in the study?

We were interested in the effect of music on the incidence of delirium in the neurosurgical population. We did not want to restrict to a certain population as a psychiatric background can be present also in neurosurgical patients. By excluding certain patients, we would bias our results, by analyzing a non-representative cohort, and limit the generalizability of our results. Moreover, excluding patients with higher risk of delirium would influence the incidence rate in the control group affecting our sample size calculation.

How were data on quality of life collected? Information on this should be added to the application section of the research.

Quality of life was assessed with the EQ-5D and EORTC QLQ – BN-20 questionnaires and a 100-likert scale. We have added this information in the methods section

How was the study protocol applied in patients undergoing emergency surgery? Was there enough time?

We were interested also in emergency cases as high urgency surgery is inherent to the neurosurgical population. Unfortunately, we were unable to include many emergency operations due to the informed consent and research intervention procedures. Therefore only 6 patients (3%) had emergency operations of which 2 patients with an operation indication within 72 hours, 3 patients within 24 hours and 1 within 6 hours. For the latter we chose not to assess the baseline questionnaires before surgery.

Why was no further analysis performed with patients who used alcohol and had a psychiatric history? Since these are important factors affecting delirium, it is necessary to examine whether they contributed to the development of delirium.

It is indeed important to know which patients develop delirium. We have conducted and published a retrospective cohort study to evaluate the risk factors contributing to post-operative delirium (*Kappen PR, Kappen HJ, Dirven CMF, Klimek M, Jeekel J, Andrinopoulou ER, et al. Postoperative Delirium After Intracranial Surgery: A Retrospective Cohort Study. World Neurosurg. 2023.*). We did not further analyze these correlations within this trial as this was not the scope of this trial.

References should be written in a certain writing format, all references should be re-examined in terms of spelling rules. References before 2010 should be removed from the references list

We have changed all the references towards the same writing style. Moreover, all the references before 2010 have been removed from the manuscript.

REVIEWER 3

Dr. Kazushi Maruo

Comments to the Author:

The statistical part of this manuscript seems to be generally acceptable, but only the following point should be considered.

For linear mixed-effects models, was heteroscedasticity between time points taken into account?

When comparing between groups at each time point, the covariance structure should be unstructured or, if an otherwise simple structure was used, sandwich variance should be used.

We would like to thank reviewer dr. Kazushi Maruo (reviewer 3) for critically revising our manuscript. We feel this input has significantly contributed to the quality of our paper.

As suggested by the reviewer, we have now accounted for heteroscedasticity by visually observing the plot and testing the model with the Levenetest. In case of heteroscedastic data we applied a log transformation and continued with this model.

See below an example of an heteroscedastic model with log transformation.

Without log transformation. The residuals fan out over time.

Levene's Test for Homogeneity of Variance (center = median)

Df F value Pr(>F)

group 1 16.663 5.273e-05 ***

455

With log transformation, the residuals spread keeps constant over time.

Levene's Test for Homogeneity of Variance (center = median)

Df F value Pr(>F)

group 1 0.7356 0.3915

455

Moreover, we changed the variance structure from compound symmetry (random = ~1|ID) to an unstructured covariance matrix (random = ~hrv_occasion|ID) as suggested by Brice Ozenne in 'Fitting linear mixed models in R'. This has changed some of our results which we have adjusted in the results section.

VERSION 2 – REVIEW

REVIEWER	Litofsky , Scott University of Missouri, Division of Neurological Surgery , University of Missouri School of Medicine , Columbia , MO , USA.
REVIEW RETURNED	31-Mar-2023

GENERAL COMMENTS	The authors have satisfactorily addressed the reviewers' comments.
--

REVIEWER	Kose, Gulsah Mugla Sitki Kocman Universitesi
REVIEW RETURNED	09-Apr-2023

GENERAL COMMENTS	Congratulations Manusript got better after revision
--